# IDBD-Based Beamforming Algorithm for Improving the Performance of Phased Array Radar in Nonstationary Environments

**DOI:** 10.3390/s23063211

**Published:** 2023-03-17

**Authors:** Shihan Wang, Tao Chen, Hongjian Wang

**Affiliations:** 1National Space Science Center, Chinese Academy of Sciences, Beijing 100190, China; 2School of Earth and Planetary Sciences, University of Chinese Academy of Sciences, Beijing 100049, China

**Keywords:** phased array radar, beamforming, IDBD algorithm, nonstationary environment

## Abstract

Adaptive array processing technology for a phased array radar is usually based on the assumption of a stationary environment; however, in real-world scenarios, nonstationary interference and noise deteriorate the performance of the traditional gradient descent algorithm, in which the learning rate of the tap weights is fixed, leading to errors in the beam pattern and a reduced output signal-to-noise ratio (SNR). In this paper, we use the incremental delta-bar-delta (IDBD) algorithm, which has been widely used for system identification problems in nonstationary environments, to control the time-varying learning rates of the tap weights. The designed iteration formula for the learning rate ensures that the tap weights adaptively track the Wiener solution. The results of numerical simulations show that in a nonstationary environment, the traditional gradient descent algorithm with a fixed learning rate has a distorted beam pattern and reduced output SNR; however, the IDBD-based beamforming algorithm, in which a secondary control mechanism is used to adaptively update the learning rates, showed a similar beam pattern and output SNR to a traditional beamformer in a Gaussian white noise background; that is, the main beam and null satisfied the pointing constraints, and the optimal output SNR was obtained. Although the proposed algorithm contains a matrix inversion operation, which has considerable computational complexity, this operation could be replaced by the Levinson–Durbin iteration due to the Toeplitz characteristic of the matrix; therefore, the computational complexity could be decreased to O(*n*), so additional computing resources are not required. Moreover, according to some intuitive interpretations, the reliability and stability of the algorithm are guaranteed.

## 1. Introduction

Phased array radar, in which the main beam and null are pointed toward the expected direction and beam scanning is realized by weighting the amplitudes and phases of each array element [1,2], plays an important role in gain control, resolution enhancement, and anti-jamming maximization [3,4] and has been widely used in radar, navigation, sonar, and communication [5,6,7,8]. Beamforming, which is one of the key techniques for phased array radar, allows users to adjust the direction of the main beam and nulls and, therefore, perform target detection, tracking, and interference suppression [4,9,10,11]. Thus, beamforming has attracted considerable attention from researchers.

Several studies have been performed on adaptive signal processing. Traditional algorithms, such as the minimum variance distortionless response (MVDR), linearly constrained minimum variance (LCMV), and frequency invariant beamforming (FIB) for wideband phased array radar, are usually based on the assumption of a stationary environment with fixed error performance surfaces [12]. In this case, an acceptable performance could be obtained by using a traditional gradient descent algorithm with a fixed learning rate. However, in practice, many environments are nonstationary [12,13]. In these nonstationary environments, the probability density function is non-Gaussian and the autocorrelation function and power spectra are time-varying, which might result in distortions in the beam pattern and deterioration in the output signal-to-noise ratio (SNR) if a fixed learning rate is used to update the tap weights [12].

Several studies have investigated target tracking, detection system identification, and pattern classification in nonstationary environments [14,15,16,17,18]. However, few studies have investigated beamforming in nonstationary environments. In [19], L. R. Zhang et al. proposed an adaptive beamforming algorithm to address beam pattern distortion due to coloured noise. In [20], fractional lower-order moment theory was applied to improve the performance of spatial and Doppler target localization. Ref. [21] studied an adaptive array processing method in a non-Gaussian environment by replacing the Gaussian data assumption with a multivariate elliptically contoured distribution. In [22], statistics and physics were used to analyse the eigenvalues of the sample covariance matrix, which might influence the performance of a beamformer.

In this study, the incremental delta-bar-delta (IDBD) algorithm, in which the learning rates of the tap weights of each element are different and time-varying [16,17,18] to track the continuously time-varying position of the optimal solution, is used to address the beamforming problem in a nonstationary environment. Moreover, the performance of the IDBD-based beamforming algorithm is compared with that of a fixed learning rate beamformer. Furthermore, the reliability, stability, and computational complexity of the proposed algorithm are analysed intuitively.

The remainder of this paper is organized as follows. In Section 2, the signal and system model of the narrowband beamformer are reviewed. In Section 3, the constrained cost function is established via Frost’s algorithm. Then, based on the IDBD algorithm, the iteration formula for each tap weight is summarized, and the reliability, stability, and computational complexity of the proposed algorithm are analysed intuitively. In Section 4, the numerical simulation results of the IDBD-based beamforming algorithm and those of a beamformer with a fixed learning rate are compared. Finally, the conclusions and some discussions are presented in Section 5.

## 2. Signals and Systems Modelling

Suppose that the signals of interest are narrowband; then, these signals can be denoted by their steering vectors. Without loss of generality, consider a uniform linear array (ULA) consisting of M omnidirectional antennas followed by M tap weights. The structure of the proposed narrowband beamformer is shown in Figure 1.

Suppose we have K signal sources spanning a column vector **s** = [s_0_(t), s_1_(t) … s_K−1_(t)]^T^ with the direction of arrival (DOA) vector **θ** = [θ_0_, θ_1_ … θ_K−1_]^T^. The steering matrix of these K signal sources can be written as **A** = [**a**_0_, **a**_1_ … **a**_K−1_]**^T^**, where
(1)$$a_{k}=\left[1,\exp\left(-j\frac{2\pi }{\lambda }d\sin\theta_{k}\right),\cdots \right.{\left.\exp\left(-j\frac{2\pi }{\lambda }(M-1)d\sin\theta_{k}\right)\right]}^{T}$$
is the steering vector of the k-th signal source. Then, the vector spanned by the signals received by each antenna can be written as
(2)x=As+n+v
where **x** = [x_0_, x_1_ … x_M−1_]^T^, **n** = [n_0_, n_1_ … n_M−1_]^T^ and **v** = [v_0_, v_1_ … v_M−1_]^T^ denote the vectors spanned by the total signals, AWGN, and nonstationary noise received by each antenna.

After the beamformer processes these signals, the output signal can be written as
(3)$$y=w^{H}x$$
where **w** = [w_0_, w_1_ … w_M−1_]^T^ denotes the vector spanned by the tap weights. Thus, the total output power of the beamformer is
(4)$$P_{\mathrm{total}}=E{\left|y\right|}^{2}=E\left(w^{H}xx^{H}w\right)=w^{H}Rw$$
where **R** = E(**xx**^H^) is the autocorrelation matrix of the input signal. According to the LCMV criterion, in which the direction of the main beam and null are guaranteed and the total output power is minimized, the objective function can be written as
(5)$$\begin{array}{c} \hat{w}=\arg\min w^{H}Rw\\ s.t.\;A^{H}w=f \end{array}$$

According to this formula, we can determine the optimal tap weights analytically using the Lagrange multiplier method as follows:(6)$$w_{\mathrm{opt}}=R^{-1}A{\left(A^{H}RA\right)}^{-1}f$$

## 3. IDBD-Based Beamformer

Equation (6) includes two matrix inverse operations, which are computationally complex. To reduce the complexity of this equation, which cannot be applied to higher-dimensional arrays, we determine the optimal tap weights iteratively using gradient descent. In this constrained optimization problem, Frost [23] noted that the cost function in the n-th iteration should be formulated via the Lagrange multiplier, namely,
(7)$$L(n)=w^{H}(n)Rw(n)+\lambda^{H}(n)\left(A^{H}w(n)-f\right)+\lambda^{T}(n)\left(A^{T}w^{*}(n)-f^{*}\right)$$
where **λ**(*n*) is an M-by-1 vector that denotes the Lagrange multiplier in the n-th iteration and the superscript * denotes complex conjugate. Through Wirtinger calculus, the gradient of the cost function L with respect to the tap weights is
(8)$$g(n)=\frac{\partial L(n)}{\partial w^{*}(n)}=Rw(n)+A\lambda (n)$$

In traditional algorithms, each tap weight is updated using the same time-invariant learning rate ε, namely,
(9)w(n+1)=w(n)−εg(n)

However, in the IDBD algorithm, each tap weight is updated according to different time-varying learning rates, which can be expressed as
(10)w(n+1)=w(n)−Ξ(n)g(n)
where **Ξ**(*n*) = diag(ε0(*n*), ε1(*n*)…εM−1(*n*)) denotes a diagonal matrix spanned by the learning rate of each tap weight in the n-th adaptation cycle.

To satisfy the constraint described in Equation (6) in each adaptation cycle, the parameter λ(*n*) needs to be adjusted as follows:(11)$$A^{H}w(n+1)=A^{H}w(n)-A^{H}\Xi (n)g(n)=f,\forall n$$

Substituting Equation (8) into Equation (11), we can analytically solve the Lagrange multiplier in each iteration cycle as follows:(12)$$\lambda (n)={\left(A^{H}\Xi (n)A\right)}^{-1}\left(A^{H}w(n)-A^{H}\Xi (n)Rw(n)-f\right)$$

Thus, the relationship between the cost function and the time-varying learning rate can be obtained; therefore, **Ξ**(*n*) can be updated adaptively via the block diagram shown in Figure 2.

Equation (12) shows that the solution of the Lagrange multiplier contains a matrix inversion operation, which has considerable computational complexity; however, due to the Toeplitz characteristic of the matrix **A**^H^**Ξ**(*n*)**A**, the Levinson–Durbin iteration could be used to solve **λ**. Because this operation has a computational complexity of O(*n*), additional computing resources are not required [12].

The learning rates are controlled by an adaptable memory parameter, denoted as *β*, and the relationship between the two parameters in the system identification problem is formulated as [12]
(13)$$\varepsilon_{m}(n)=\exp\left(\beta_{m}(n)\right)$$

However, the beamforming problem includes several complex operations; thus, to ensure that the learning rates are always real numbers, in this paper, we rewrite Equation (13) as
(14)$$\varepsilon_{m}(n)=\exp\left(\beta_{m}(n)+\beta_{m}^{*}(n)\right)$$

Thus, the cost function is formulated as a function of the adaptable memory parameter, allowing us to control the learning rate matrix **Ξ**(*n*) by updating *β* via gradient descent with a fixed learning rate κ. Next, we show the mathematical derivations.

The differentiation of the cost function with respect to the m-th tap weight in the n-th adaptation cycle is
(15)$$\begin{array}{c} g_{m}(n)=\frac{\partial L(n)}{\partial w_{m}^{*}(n)}=\sum_{l=0}^{M-1}\left(R_{ml}w_{l}(n)+A_{ml}\lambda_{l}(n)\right)\\ =R(m,:)w(n)+A(m,:)\lambda (n) \end{array}$$
where **A**(m,:) and **R**(m,:) denote the m-th row of matrix **A** and **R,** respectively. Then, the updated formula for the m-th tap weight can be expressed as
(16)$$w_{m}(n+1)=w_{m}(n)-\varepsilon_{m}(n)\left(R(m,:)w(n)+A(m,:)\lambda (n)\right)$$

Based on this formula, the gradient of the cost function with respect to the adaptable memory parameter can be determined using the chain rule for differentiation, namely,
(17)$$\frac{\partial L(n)}{\partial \beta_{m}^{*}}=\frac{\partial L(n)}{\partial w_{m}^{*}(n)}\cdot \frac{\partial w_{m}^{*}(n)}{\partial \beta_{m}^{*}}=g_{m}(n)h_{m}(n)$$
where hm(*n*) denotes the differentiation of the m-th tap weight with respect to the m-th adaptable memory parameter in the n-th adaptation cycle. This equation is difficult to solve analytically. However, Equations (14) and (16) suggest the following relationship:(18)$$\begin{array}{c} h_{m}(n+1)=\frac{\partial w_{m}^{*}(n+1)}{\partial \beta_{m}^{*}}=\\ \frac{\partial }{\partial \beta_{m}^{*}}{\left(w_{m}(n)-\varepsilon_{m}(n)g_{m}(n)\right)}^{*}=\\ h_{m}(n)-\varepsilon_{m}(n)g_{m}^{*}(n)-\varepsilon_{m}(n)\frac{\partial g_{m}^{*}(n)}{\partial \beta_{m}^{*}} \end{array}$$

Thus, we can determine *h*_m_(*n*) iteratively.

Substituting Equations (12) and (15) into Equation (18), and in order to make the mathematical form more convenient, let auxiliary variables
(19)$$\begin{array}{c} C(n)={\left(A^{T}\Xi (n)A^{*}\right)}^{-1}\\ D(n)=I-\Xi (n)R^{*} \end{array}$$

Then, the differentiation of *g*_m_(*n*) with respect to *β*_m_(*n*) can be written as
(20)$$\begin{array}{c} \frac{\partial g_{m}^{*}(n)}{\partial \beta_{m}^{*}}=\\ \frac{\partial }{\partial \beta_{m}^{*}}{\left(R(m,:)w(n)+A(m,:)\lambda (n)\right)}^{*}=\\ R_{mm}h_{m}(n)+\\ A^{*}(m,:)\frac{\partial }{\partial \beta_{m}^{*}}\left(C(n)\left(A^{T}D(n)w^{*}(n)-f\right)\right) \end{array}$$

Moreover, the differentiation in the fourth line of Equation (20) can be expanded as three terms:(21)$$\begin{array}{c} \frac{\partial }{\partial \beta_{m}}\left(C(n)\left(A^{T}D(n)w^{*}(n)-f\right)\right)=\\ \frac{\partial C(n)}{\partial \beta_{m}}\left(A^{T}D(n)w^{*}(n)-f\right)-\\ C(n)A^{T}\frac{\partial \Xi (n)}{\partial \beta_{m}}R^{*}w^{*}(n)+\\ C(n)A^{T}D(n)\frac{\partial w^{*}(n)}{\partial \beta_{m}} \end{array}$$

Using the matrix differential formula, we obtain
(22)$$\frac{\partial \Xi (n)}{\partial \beta_{m}}=\varepsilon_{m}(n)\left[\begin{array}{ccc} O & & \\ & 1 & \\ & & O \end{array}\right]$$
(23)$$\begin{array}{c} \frac{\partial C}{\partial \beta_{m}}=-CA^{T}\frac{\partial \Xi (n)}{\partial \beta_{m}}A^{*}C=\\ -\varepsilon_{m}(n)C{\left(A(m,:)\right)}^{T}A^{*}(m,:)C \end{array}$$
(24)$$\frac{\partial w^{*}(n)}{\partial \beta_{m}}={\left[0,h_{m}(n),0\right]}^{T}$$

Thus, Equation (20) can be further simplified as
(25)$$\begin{array}{c} \frac{\partial }{\partial \beta_{m}}\left(C(n)\left(A^{T}D(n)w^{*}(n)-f\right)\right)=\\ -\varepsilon_{m}(n)C{\left(A(m,:)\right)}^{T}A^{*}(m,:)\\ C\left(A^{T}D(n)w^{*}(n)-f\right)\\ -\varepsilon_{m}(n)C{\left(A(m,:)\right)}^{T}R^{*}(m,:)w^{*}(n)\\ +h_{m}(n)CA^{T}D(:,m) \end{array}$$
where **D**(:,m) denotes the m-th column of matrix **D**. Substituting Equations (20) and (24) into Equation (18), the gradient of the cost function with respect to the adaptable memory parameter in each iteration cycle can be obtained, and the updated formula for *β*m(*n*) can be written as
(26)$$\beta_{m}(n+1)=\beta_{m}(n)+\kappa \frac{\partial L}{\partial \beta_{m}^{*}(n)}$$
where κ is a fixed learning rate for updating *β*m(*n*). Thus, by substituting Equation (26) into Equation (14), the time-varying learning rate for the m-th tap weight in the n-th adaptation cycle can be adaptively controlled. Moreover, Equations (18)–(26) show that by choosing this relationship between ε and *β*, *β* becomes proportional to the product of the current weight change and the trace of recent weight changes, as shown in Equation (14). Therefore, after several iterations, the overall change in *β* becomes proportional to the correlation between the current and recent weight changes, which suggests that the previous steps were larger if the current learning rate is positively correlated with past learning rates, and vice versa. According to this intuitive statement, the iteration of *β* is controlled by previous states, indicating that the update direction of the tap weights is corrected adaptively; thus, the proposed algorithm is reliable and stable. A similar analysis is provided in [18].

Summarizing the above mathematical derivations, the pseudocode for the IDBD-based beamformer is presented in Algorithm 1.
**Algorithm 1** Summary of the IDBD-based beamformer.**begin initialize** w(0),Ξ(0),h(0),m←0,n←0**input** A,κ,N**while** n≤N**compute** C(n),D(n),λ(n) with Equations (12) and (19)**compute** $g_{m}(n)$ with Equation (15)         **compute** $\partial g_{m}^{*}(n)/\partial \beta_{m}^{*}$ with Equation (25)         **compute** $\partial L(n)/\partial \beta_{m}^{*}$ with Equation (17)         **do**$\;h_{m}(n+1)\leftarrow h_{m}(n)-\varepsilon_{m}(n)g_{m}^{*}(n)-\varepsilon_{m}(n)\frac{\partial g_{m}^{*}(n)}{\partial \beta_{m}^{*}}$$\;\beta_{m}(n+1)\leftarrow \beta_{m}(n)+\kappa \frac{\partial L}{\partial \beta_{m}^{*}(n)}$                εm(n+1)←exp2Reβm(n+1)                w(n+1)←w(n)−Ξ(n)g(n)                m←m+1**until** m=M−1n←n+1**end**

## 4. Numerical Simulation

In the numerical simulation of the algorithm, we chose a ULA with 16 antenna elements and an element spacing of 1/2-times the wavelength of the carrier wave. Suppose that the target signal is a linear frequency-modulated (LFM) signal with a pulse width of 10 μs and a bandwidth of 12 MHz, and that the power and DOA of the target signal are 0 dB and 30°, respectively. We choose three interference signals with powers of 5 dB: sine waves with frequencies of 8 GHz, 9 GHz, and 9.5 GHz and incident DOAs of −60°, −30°, and 0°, respectively. The powers of the AWGN and nonstationary signals are 3 dB and 15 dB, respectively.

The nonstationary noise was randomly chosen from the heavy sine, bumps, and Doppler signals in each Monte Carlo experiment. The waveform and autocorrelation functions of these nonstationary noises are shown in Figure 3. The figures show that for each time delay, the autocorrelation functions of these signals vary with the initial observation time, which indicates the nonstationary property of these signals.

The autocorrelation matrix of the received signal x is computed by replacing the ensemble average with an arithmetic average. The tap weights are initialized randomly, and the diagonal matrix spanned by the time-varying learning rates is initialized to 0.005-times the unitary matrix. Furthermore, to compare the performance of the IDBD-based beamformer with that of a beamformer with a fixed learning rate of 0.005, we studied the performance of the two algorithms in a stationary environment with an AWGN signal with the same power. Figure 4 shows the average results of 100 Monte Carlo experiments, and in each of the experiments, the snapshot is 128.

In Figure 4, the blue dotted line, cyan circle, red dotted line, and pink dotted line denote the performance of the IDBD-based beamformer in nonstationary and stationary environments and the performance of the beamformer with a fixed learning rate in nonstationary and stationary environments, respectively. We plotted the SNR curve of the traditional beamformer in a nonstationary environment in a separate graph to ensure that the changes in the three SNR curves could be shown clearly.

Figure 4 shows that in a stationary environment, the IDBD-based beamformer and fixed learning rate beamformer show approximately the same performance, namely, the main beam and null point toward the expected direction and the optimal SNRs are output. However, in a nonstationary environment, although the nulls satisfied the pointing constraint for the beamformer with a fixed learning rate, the direction of the main beam was distorted, and the output SNR decreased dramatically as the number of iterations increased. This might be because when the surface of the cost function L moves in the weight space due to the time-varying property of auto-correlation matrix R, we still update the weight vector along a fixed direction. Therefore, the algorithm cannot converge to the optimal solution and result in the distortion of the radiation pattern and deterioration of the output SNR.

In contrast, for the IDBD-based beamformer, the beam pattern and SNR showed similar performance to the IDBD-based beamformer in a stationary environment. Moreover, Figure 4b shows that the convergence rate of the IDBD-based beamforming algorithm is as fast as the convergence rates of both beamformers in a stationary environment. A similar result was discussed in [17], revealing the superiority of the IDBD-based beamforming algorithm.

In conclusion, the traditional beamforming algorithm with a fixed learning rate and the IDBD-based beamforming algorithm both showed acceptable performance in a stationary environment; that is, the main beam pointed toward the target direction, the null point pointed toward the interference, and the optimal output SNR was obtained. However, in an environment with high-power nonstationary noise, the traditional beamformer suffers from beam pattern distortions and a reduced output SNR, and an IDBD-based beamformer is needed to address these issues.

## 5. Conclusions and Discussions

Several studies have shown that in nonstationary environments, algorithms with fixed learning rates may fail to converge to the optimal solution, resulting in a reduced performance [20,21,22], which is reflected in beam pattern distortion and a decreased SNR in beamforming problems. Therefore, for a beamformer to achieve its optimal tracking capability, the beamformer must first pass from transient mode to steady-state mode, and the filter′s free parameters must be continuously adjusted [12]. By introducing time-varying learning rates for each weight, the algorithm can track statistical changes in the environment in real time.

We considered an LFM target signal and compared the performance of the IDBD-based beamforming algorithm with that of a beamformer with a fixed learning rate in stationary and nonstationary environments. The results indicate that the IDBD-based beamformer has several advantages.

First, in contrast with the beamformer with a fixed learning rate, which had beam pattern distortion and a decreased SNR in nonstationary environments, the IDBD-based beamforming algorithm performs similarly to a traditional beamformer in a stationary environment; that is, the main beam and null pointed toward the expected direction, and the optimal output SNR was obtained.

Second, in a nonstationary environment, the convergence rate of the IDBD-based beamforming algorithm was as fast as that of the beamformer with a fixed learning rate in a stationary environment, which indicates that the introduction of the adaptable memory parameter does not deteriorate the rate of convergence.

Finally, Equations (12), (19) and (25) show that in the IDBD-based beamforming algorithm, although each tap weight adaptation cycle includes a matrix inversion operation, the Toeplitz characteristic of the matrix **A**^H^**Ξ**(*n*)**A** indicates that this problem can be solved using the Levinson–Durbin iteration, which has a computational complexity of O(*n*). This complexity is the same as that of the beamforming algorithm with a fixed learning rate; thus, additional computing resources are not required.

Due to these advantages, we suggest that the IDBD-based beamformer should be implemented in extremely harsh electromagnetic environments, such as magnetic storms, in which the background electromagnetic noises showed similar characteristics of the simulated environment used in this paper, namely non-Gaussian and nonstationary characteristics [24].

## Figures and Tables

**Figure 1 sensors-23-03211-f001:**
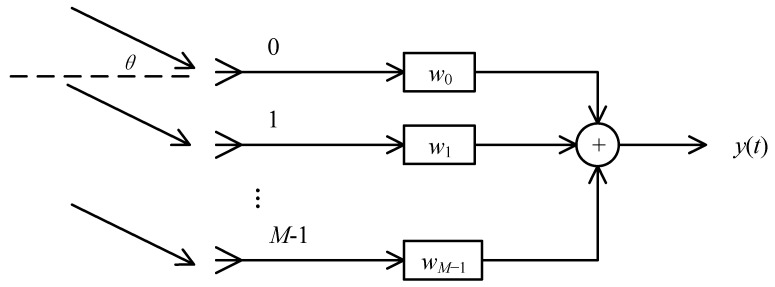
Structure of the narrowband beamformer.

**Figure 2 sensors-23-03211-f002:**
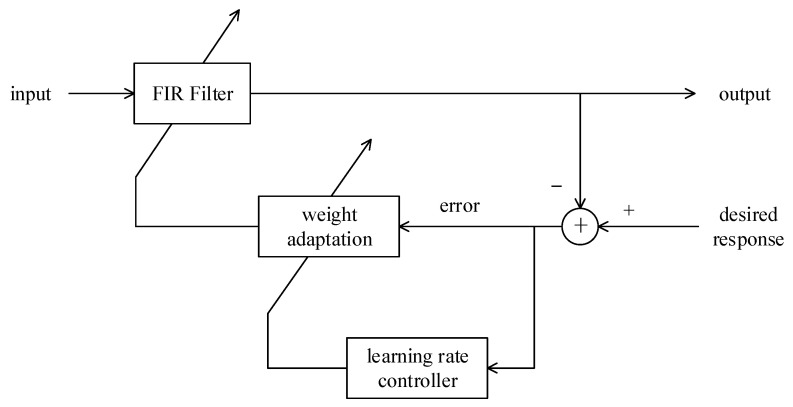
Block diagram of the IDBD algorithm.

**Figure 3 sensors-23-03211-f003:**
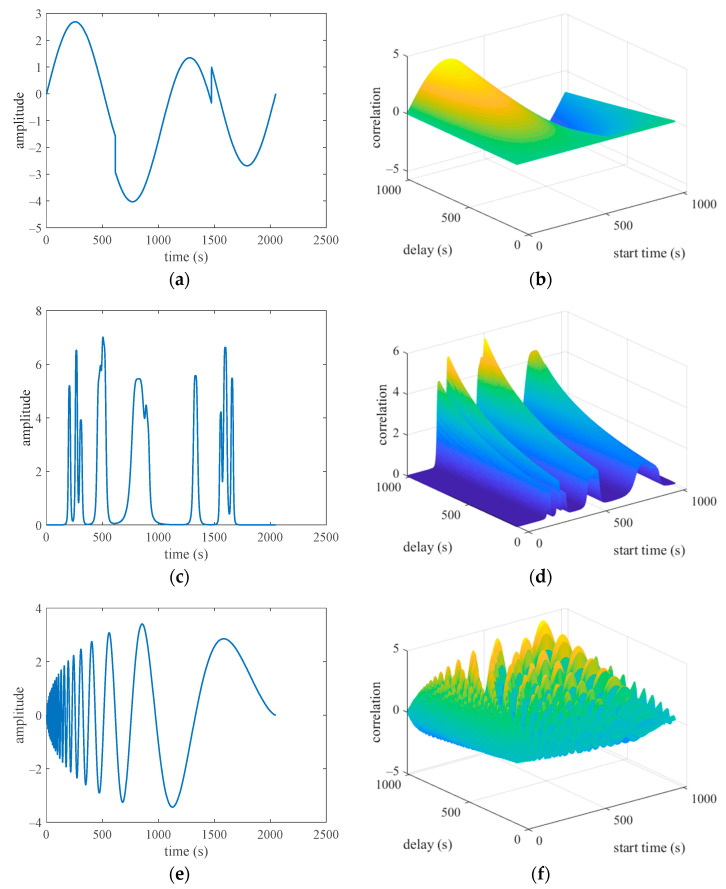
Common nonstationary signal waveforms and their autocorrelation functions (panels (**a**,**c**,**e**) are waveform of the heavy sine signal, bumps signal and Doppler signal, and panels (**b**,**d**,**f**) are autocorrelation function of the heavy sine signal, bumps signal and Doppler signal).

**Figure 4 sensors-23-03211-f004:**
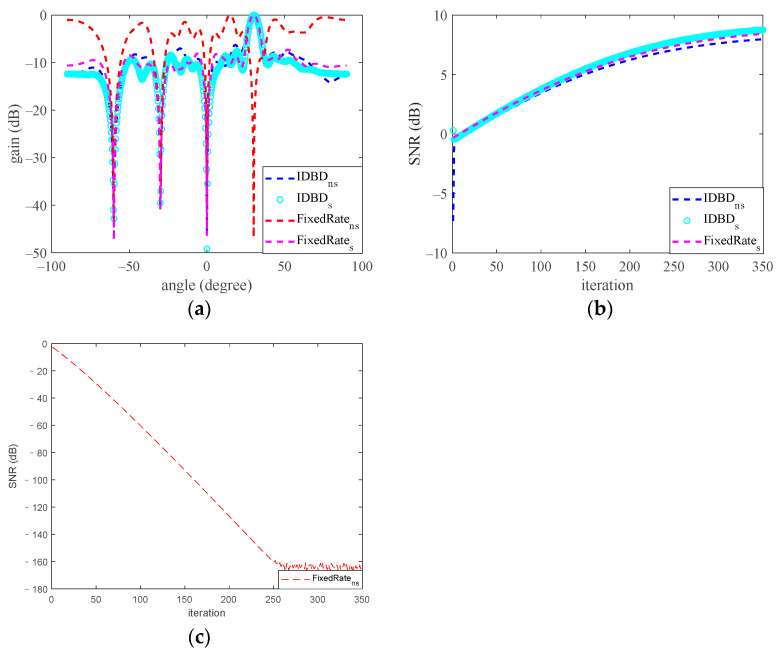
Comparison of the performance of the two beamforming algorithms in stationary and nonstationary environments (panel (**a**) is beam pattern, panel (**b**) indicates output SNR of the IDBD-based beamformer in stationary and nonstationary environments and the fixed learning rate beamformer in a stationary environment, and panel (**c**) represents output SNR of the fixed learning rate beamformer in a nonstationary environment).

## Data Availability

There are no new data.

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
