# Peer review of "IDBD-Based Beamforming Algorithm for Improving the Performance of Phased Array Radar in Nonstationary Environments"

_sensors, 2023, doi:10.3390/s23063211_

Round 1
Reviewer 1 Report
In this paper, the authors proposed beamforming method for phase array radar in non-stationary environments. They also conducted simulations to support their conclusion about their proposal. However, I have the following comments.
1) In equations, there are some notations such as A(m,:). I guess it means the m-th row in matrix A, but it is better to define this notation.
2) The matrix notation should be in bold font like the notation in equations.
3) The font size of equations is too large, while the sizes of Fig. 1 and 2 are too small.
4) It is better to explain why the output SNR of fixed rate goes below -150 dB, which is not a reasonable SNR value.
Author Response
Reviewer1:
Suggestion1:
In equations, there are some notations such as A(m,:). I guess it means the m-th row in matrix A, but it is better to define this notation.
Reply1:
Thanks for your reviewing, this is our negligence and we have added the explanation of the notations A(m,:) in the paper and marked them red. In order to your convenience, we copied these explanations as follows:
Where A(m,:) and R(m,:) denotes the m-th row of matrix A and R respectively
Where D(:,m) denotes the m-th column of matrix D
which can be seen in line 158 and 184 respectively in the revised paper
Suggestion2:
The matrix notation should be in bold font like the notation in equations.
Reply2:
Thanks for your careful review, now, all the matrix notation are in bold font.
Suggestion3:
The font size of equations is too large, while the sizes of Fig. 1 and 2 are too small
Reply3:
Thanks for your suggestion, we have enlarged Fig. 1 and 2 and changed the font size of equations.
Suggestion4:
It is better to explain why the output SNR of fixed rate goes below -150 dB, which is not a reasonable SNR value
Reply4:
Thanks for your advice, we have made an intuitive explanation on the deterioration of the output SNR of beamformer with fixed learning rate, which is marked red in the revised paper from line 252 to 257 and in order for your convenience, we also copied it as follows:
This might because that when the surface of the cost function L moving in the weight space due to the time-varying property of auto-correlation matrix R, we still update the weight vector along a fixed direction. Therefore, the algorithm cannot converge to the optimal solution and result in the distortion of the radiation pattern and deterioration of the output SNR.

Reviewer 2 Report
1) This paper covers the improvement of beamforming algorithm for phased array radar in nonstationary environment better than traditional methods using incremental delta-bar-delta algorithm (IDBD). To reduce the complexity of equation to use higher-dimensional arrays due to inverse matrix operator, the optimal tap weights was determined. So this paper used IDBD-based beamforming algorithm strong in extremely harsh nonstationary environment. It is recommended to revise the title of the paper to better represent these specific contributions.
2) In Section 5, IDBD-based algorithms are expected to be more robust than fixed learning rates in non-stationary environments such as magnetic storms. Then, is the simulated environment used in the thesis similar to a magnetic storm? If so, it seems that a sufficient explanation for this should be added.
3) It would be better if experimental verification of the IDBD-based algorithm followed by computational research. If so, could you describe your future research?
Author Response
Suggestion1:
This paper covers the improvement of beamforming algorithm for phased array radar in nonstationary environment better than traditional methods using incremental delta-bar-delta algorithm (IDBD). To reduce the complexity of equation to use higher-dimensional arrays due to inverse matrix operator, the optimal tap weights was determined. So this paper used IDBD-based beamforming algorithm strong in extremely harsh nonstationary environment. It is recommended to revise the title of the paper to better represent these specific contributions
Reply1:
Thanks for your suggestion and in order to better represent these specific contributions of this paper, we changed the title into: “IDBD Based Beamforming Algorithm for Improving the Performance of Phased Array Radar in Nonstationary Environments”
Suggestion2:
In Section 5, IDBD-based algorithms are expected to be more robust than fixed learning rates in non-stationary environments such as magnetic storms. Then, is the simulated environment used in the thesis similar to a magnetic storm? If so, it seems that a sufficient explanation for this should be added.
Reply2:
Thanks for your suggestion. in reference [24], the authors have studied the probability density functions, auto-correlation functions and power spectrum of the background electromagnetic noises under magnetic storm. The results show that when a magnetic storm occur, the background electromagnetic noises showed non-Gaussian, non-stationary and time-varying power spectrum, which is similar to the simulated environment used in this paper.
In the revised paper, we added the following comments, which are marked red from line 302 to 304:
Due to these advantages, we suggest that the IDBD-based beamformer should be implemented in extremely harsh electromagnetic environments, such as magnetic storms, in which the background electromagnetic noises showed similar characteristics of the simulated environment used in this paper, namely non-Gaussian and nonstationary characteristics[24].
Suggestion3:
It would be better if experimental verification of the IDBD-based algorithm followed by computational research. If so, could you describe your future research?
Reply3:
Thanks for your suggestion. We are currently studying the DOA estimation algorithm for phased array radar under non-stationary environment, once we complete that, we will use a real radar to carry some experimental for both DOA and DBF.

Reviewer 3 Report
In the present manuscript, the IDBD algorithm is used to improve beamforming algorithm with fixed learning rate. The beam signal quality in nonstationary environments is better improved. However, the organization of the paper is somewhat chaotic. In order to improve the quality of the manuscript, the following comments should be addressed.
1. In the second paragraph of Section I, the full name of LCMV is incorrect.
2. There is some confusion in the equation arrangement. For example, some equation numbers are omitted (Eq. (1)), but other equation numbers are repeated (Eq. (20)). There is no need to start another paragraph for the parameter interpretation in line 86 (The same problem will not be duplicated).
3. What is the meaning of As in Eq. (2)? I also can't find the explanation of C(n), D(n) and I in Eq. (19). The L in Eq. (26) refers to L(n) in Eq. (7)?
4. It is mentioned in the paper that the reason for the rewriting of Eq. (13) to Eq. (14) is to ensure that the learning rate is always real number, but the implementation method is not specified. Can this be achieved just by rewriting the formula form? Does the article ignore some practical factors or adopt some theoretical approximations? Is this reasonable?
5. In the first paragraph of the fourth subsection, why is the power in dB?
6. In Fig. (1) & (2), the font size is too small; the line is too thin and unclear.
7. The comparison results of Figure 4 are somewhat messy and not intuitive enoug. It is recommended to combine Fig.4 (b) & (c).
Author Response
Suggestion1:
In the second paragraph of Section I, the full name of LCMV is incorrect
Reply1:
Many thanks to your careful review! We have corrected the full name of LCMV: “Linearly constrained minimum variance” and it have been marked red in the revised paper.
Suggestion2:
There is some confusion in the equation arrangement. For example, some equation numbers are omitted (Eq. (1)), but other equation numbers are repeated (Eq. (20)). There is no need to start another paragraph for the parameter interpretation in line 86 (The same problem will not be duplicated).
Reply2:
Thanks for your suggestion. We have corrected the equation arrangement and deleted the duplicated parameter interpretation.
Suggestion3:
What is the meaning of As in Eq. (2)? I also can't find the explanation of C(n), D(n) and I in Eq. (19). The L in Eq. (26) refers to L(n) in Eq. (7)?
Reply3:
Thanks for your question. In fact, As denotes the matrix multiplication of steering matrix A and signal vector s, which means the component of received data from the signals and interference.
The matrix C(n) and D(n) are just auxiliary variables for the convenience of mathematical form and do not have physical meanings. We have added the explanation in the revised paper, which is marked red in line 171 and 172. in order for your convenience, we also copied it as follows:
and in order to the convenient of mathematical form, let auxiliary variables
(19)
The L in Eq. (26) refers to L(n) in Eq. (7), we ignored the variable n just for mathematical convenience.
Suggestion4:
It is mentioned in the paper that the reason for the rewriting of Eq. (13) to Eq. (14) is to ensure that the learning rate is always real number, but the implementation method is not specified. Can this be achieved just by rewriting the formula form? Does the article ignore some practical factors or adopt some theoretical approximations? Is this reasonable?
Reply4:
In Eq. (14), we added the parameter β to its complex conjugate β *, therefore, it is naturally that the imaginary part of this parameter would be eliminated by this implementation.
Suggestion5:
In the first paragraph of the fourth subsection, why is the power in dB
Reply5:
Thanks for your question. In fact, we us dB as the unit of power in order to show the power of noises and interference compared with that of the target signal. That’s also the reason why we set the power of the target signal 0 dB.
Suggestion6:
In Fig. (1) & (2), the font size is too small; the line is too thin and unclear
Reply6:
Thanks for your suggestion, we have enlarged Fig. 1 and 2. Now they should be more clear
Suggestion7:
The comparison results of Figure 4 are somewhat messy and not intuitive enough. It is recommended to combine Fig.4 (b) & (c)
Reply7:
Thanks for your suggestion. We have set the curves bolder in Fig. 4 (a) and it should be more clear and intuitive
We did not combine Fig.4 (b) & (c) because it is decreased too sharply of the output SNR of traditional beamformer under non-stationary environment that if we combine Fig.4 (b) & (c), the trend of the learning curve in (c) cannot be shown clearly.

Round 2
Reviewer 2 Report
Thank you for your response.